# Linking Passion for Work and Emotional Exhaustion in Indonesian Firefighters: The Role of Work–Family Conflict

**DOI:** 10.3390/ijerph192214629

**Published:** 2022-11-08

**Authors:** Jovi Sulistiawan, Massoud Moslehpour, Pei-Kuan Lin

**Affiliations:** 1Department of Business Administration, Asia Management College, Asia University, Taichung 41354, Taiwan; 2Department of Management, Faculty of Economics and Business, Universitas Airlangga, Surabaya 60115, Indonesia; 3Department of Management, California State University, San Bernardino, CA 92407, USA

**Keywords:** the dualistic model of passion, emotional exhaustion, firefighters, passion for work, work–family conflict

## Abstract

This study employs a theoretical and comprehensive framework for investigating the relationship between passion for work, work–family conflict, and emotional exhaustion. Drawing from the dualistic model of passion, we posited that passion could provoke negative feelings, leading to strict determination and inhibiting the attainment of an effective, balanced life. However, there is little empirical evidence to support the dualistic model of passion’s notion that passion either can assist employees in balancing their various life responsibilities or impede such a balance. The purposes of this study are threefold: first, to investigate the impact of passion for work on work–family conflict; second, to examine the relationship between work–family conflict and emotional exhaustion; and third, to clarify the mediating process of work–family conflict in the relationship between passion for work and emotional exhaustion. A cross-sectional survey was employed to gather data from Indonesian firefighters (*n* = 398). PLS-SEM was utilized to test the proposed hypotheses. Our results revealed that obsessive passion negatively affects emotional exhaustion. The underlying reason for this result is due to self-conceptions based on community expectations, receiving help from others to solve problems, and improving well-being even when problems arise. Our results suggest that organizations encourage employees’ harmonious development, providing employees with skills necessary to deal with challenging situations and providing a family-supportive policy.

## 1. Introduction

As members of the emergency responders, firefighters are repeatedly exposed to life-threatening and incredibly stressful situations. Multiple studies have demonstrated that firefighters suffer from high levels of emotional exhaustion—the primary symptom of burnout [1,2]. Prior studies have investigated the causes of emotional exhaustion, such as excessive job demands, a lack of social support, interpersonal conflict, etc. However, Landay et al. [3] argued that a strong inclination toward work that people love, which makes them invest more time and energy in one activity, becomes the potential cause of emotional exhaustion. This strong inclination toward work is called a passion for work. Despite the fact that passion promotes willing involvement, it may support both benefits and disadvantages [4].

In recent years, passion for work has gained considerable academic and managerial attention [3,5,6]. A decade ago, the dualistic model of passion (DMP) proposed that there are two types of passion: harmonious and obsessive passion [7,8]. According to the DMP, having an obsessive passion causes conflict between the passionate activity and other aspects of one’s life. A harmonious passion allows for easier integration and, thus, less friction. However, despite the dualistic model of passion’s assertion that harmonious work passion helps employees to manage their numerous life roles, whereas obsessive work passion interferes with such balance, there is little empirical evidence to support this claim [9].

Chen et al. [10] asserted that earlier studies only addressed passion as a unidimensional construct and were unable to explain why two employees can have the same level of passion for their profession but distinct psychological results. Moreover, according to Bayraktar and Jimenez [5], equating harmonious passion with consistently favorable outcomes and obsessive passion with consistently negative effects would oversimplify the circumstances. Indeed, prior studies have revealed contradictory conclusions about whether harmonious passion is always beneficial and obsessive behavior is always detrimental [4,8,11]. These contradictory findings, especially in the case of obsessive passion, imply that additional investigation is required to examine the impacts of passion in various circumstances and a more specific domain in the near future.

In their study limitations, Landay et al. [3] advised that first responders who are likely to experience greater levels of passion, such as firefighters, would be excellent for further research on how passion influences burnout. These employment positions and obligations are among the most dangerous and physically and mentally taxing experienced by any workforce [12]. Beuchamp [13] found that firefighters considered passion the most important aspect of firefighting culture. Moreover, the job features of firefighters include dealing with physically demanding work, time constraints, and emotionally taxing events, all of which have been associated with emotional exhaustion, which is the main symptom of burnout [2]. Several prior studies in the field of occupational health have investigated burnout in the context of firefighters [2,14]. However, to our best knowledge, the research that examines burnout in the context of firefighters with the lens of a dualistic model of passion remains scarce. In order to ensure the well-being of firefighters, it is necessary to investigate how passions differently affect emotional exhaustion.

Vergauwe et al. [15] asserted that although passion stimulates motivation and provides meaning in daily life, it can also evoke negative emotions, resulting in rigid persistence, and impeding the achievement of an effective, balanced life. Passionate individuals dedicate a substantial amount of time and energy to their passion and may even give it priority status. Such a condition makes it difficult for individuals to manage and balance the demands of passion with those of other essential activities or life domains. Therefore, work–family conflict (WFC) may be unavoidable for everyone, but it seems that for employees with great passion, WFC may be a part of life [6]. Prior studies have studied the link between passion for work and WFC, verifying a positive correlation between the two constructs [6,9,16,17]. Among those prior studies, only Houlfort et al. [6] used a multidimensional approach to WFC, leaving us with an incomplete picture of how passion for work influences WFC and how WFC mediates the relationship between passion and employee well-being.

The objectives of this study are as follows. First, drawing from the dualistic model of passion, this study investigates the effect of passions on work–family conflict. Second, it investigates how work–family conflict affects emotional exhaustion. Third, it clarifies the mediating mechanism of work–family conflict in the relationship between passions and emotional exhaustion. In order to attain our objectives, we employed the PLS-SEM method in multiple ways. First, to ensure the data’s quality, we employed PLS-SEM to assess the validity and reliability of our data. Second, after ensuring validity and reliability, we used structural path analysis to test our proposed hypotheses.

Although studies related to DMP have been conducted for over a decade, several points need clarification. Thus, this study offers significant theoretical and practical implications. First, this study sheds some light on the employee well-being literature by investigating the effect of passion. Second, this study, according to our knowledge, is the first paper studying the DMP in the context of firefighters. Third, if our antecedent variables are found to be significant predictors of emotional exhaustion, they may provide targets for initiatives within the fire brigade to reduce emotional exhaustion.

### 1.1. Dualistic Model of Passion (DMP)

Vallerand et al. [7] introduce the dualistic model of passion (DMP) by using self-determination theory [18] as a theoretical lens. In DMP, passion is defined as a person’s extremely intense attraction to a particular activity that they consider critical to their existence. Indeed, the individual will devote a substantial amount of time and energy to these pursuits [6]. It can be concluded that passion is likely to arouse a person’s inherent urge to engage in a certain activity. This enthusiasm will eventually result in the individual feeling eager and committed to complete their work, resulting in positive emotions. However, Curran et al. [4] argued that passion does not necessarily result in favorable emotions. Passion can result in excessive cravings and bad emotions because of an individual’s excitement, which eventually results in pressure.

Vallerand [9] defined passion as a dualistic paradigm that consists of seven core elements that set it apart from other activities. First, passion manifests itself in specific activities or pursuits, not in more generic pursuits that are exciting to anyone. Second, passion refers to feelings of love, even intense love, for a particular activity. Third, passion manifests itself only in activities or endeavors that are extremely significant and valuable to someone. Fourth, passion is more than a sensation, it is a determination to act. Fifth, passion displays itself when an activity appears to be an integral part of an individual’s identity. Sixth, passion always engages a person’s psychology, regardless of the effort, energy, and dedication required to accomplish certain goals. Last, passion takes two opposing forms and can result in adaptive and maladaptive results [4].

The DMP classifies passion into two categories: harmonious and obsessive [6]. Harmonious passion (HP) originates from autonomous internalization [5,18]. It implies that HP is the outcome of a sustained and voluntary internalization process for a particular activity, without any pressure to perform it until it becomes an integral part of one’s self-identity [6]. Individuals who strongly invest their psychological energies in activities that have been used as a source of passion and have become absorbed into the individual’s identity can also demonstrate HP [19]. HP is able to generate a powerful driving factor to engage in voluntary activities that are loved and have been ingrained in the individual’s personality [20]. Additionally, individuals retain complete control over the activities that have developed into their life passions, even if these activities have become an integral part of their personality [19,21]. People with harmonious passions participate in their work because of some inherent qualities of the activity in the workplace (e.g., fun and challenging) [9,22]. Additionally, people who are passionate about their work are able to harmoniously integrate work with other responsibilities and elements of their lives since work becomes an autonomous or volitional part of who they are [9]. They consequently reported fewer negative emotions, such as guilt and anxiety, and more positive impacts, such as pleasure and enjoyment, when participating in activities [7]. In other words, everyone with an HP is able to decide when to engage in certain activities and when not to. Individuals have the option of stopping their engagement in activities that have evolved into their passion if these activities have a detrimental effect on their lives.

On the other hand, obsessive passion (OP) comes as a result of a controlled internalization process; in other words, both internal and external factors influence a person’s decision to internalize a particular activity [6]. The internal pressure is derived from an individual’s self-qualities, such as a high self-esteem. Similarly, the external pressure originates externally, such as social approval or a desire for a high-performance appraisal [19]. However, the person’s investment in psychological resources and affection is uncontrollable [21]. In other words, passion for the activity begins to control the individual instead of the individual controlling the activity [5]. As a result, individuals are compelled to engage in activities they perceive as their passion. This is because individuals have a deep desire to engage in enjoyable and meaningful activities [20]. Therefore, even while those with obsessive passions like what they do, the conflicting and compulsive nature of obsessive urges has an influence on their emotional experience, causing them to feel less joy and more anxiety when engaging in activities and when they are stopped from doing so [7,8]. As a result, people are prevented from completely engaging in activities and are more likely to have a performance orientation that emphasizes competing with others rather than learning and developing expertise [9]. Thus, even if they are unable to devote psychological energy to activities of their interest, they will continue to follow and carry out the activities [19,20,21].

The concept of passion is similar to other work-attitude concepts such as work engagement and workaholism. Although there are conceptual similarities, they are conceptually distinct in several aspects. Rip et al. [23] asserted that a person’s self-concept is linked to their passion for work. This is not a requirement for workers with job engagement or workaholism. Kahn [24] initially defined work engagement as maximizing the employee’s self-utilization; this terminology currently refers to an employee’s capacity to be physically, mentally, and emotionally involved and is unrelated to self-concept and identity. Work becomes a component of the employee’s identity when the job meets internal desires (supremacy, self-esteem, and social approval) coupled with outward fulfillment [25].

Spence and Robbins [26] characterized a workaholic as a person who exhibits three traits compared to others: firstly, workaholics are deeply involved in their work; secondly, they feel obliged to work due to inner pressure; and thirdly, they tend not to love what they do. Workaholics do not necessarily love or appreciate their work. This is the most fundamental distinction between the OP and workaholics. While workaholics work obsessively and excessively but do not enjoy their occupations [27], employees with strong OPs will work a lot and have difficulties separating themselves from work, but they do it because they enjoy their work. Workaholics appear to be devoted to working, whereas employees with strong OPs appear to be addicted to their specific job and the benefits it may provide. Workaholism theories have focused predominantly on the necessity for employees to feel active in general, rather than on whether they are explicitly engrossed in their work.

### 1.2. Work–Family Conflict (WFC)

Work–family conflict is defined as a type of inter-role conflict in which the demands of the work and family domains collide in some way [12,28,29]. Greenhaus and Beutell [28] classified work–family conflict into two categories: work that interferes with family (WIF), and family that interferes with work (FIW). WIF is defined as a conflict of roles between work and family caused by work interfering with family members’ respective roles [30]. The individuals believe that they are allocating all the resources available to perform the task in the workplace. These jobs and demands divert attention and energy away from an individual’s family [31]. FIW refers to a role conflict between family and work in which the family considers intervening in individual roles at work [30]. Compared to WIF, individuals perceive high demands from their family and this eventually interferes with their roles and obligations in the workplace. For instance, individuals arrive late for work due to the need to care for their children [31].

In addition to the direction of WFC (WIF or FIW), WFC can also take different forms; that is, it can be time-based, strain-based, or behavior-based [6,28]. Time-based conflict takes place when an individual’s time commitment to one role inhibits participation in another [6]. Strain-based conflict emerges when it becomes apparent that what one position entails poses a threat to another [6]. For instance, when an organization announces massive layoffs, employees are so obsessed with whether they will keep their jobs that it is challenging to be present and fully participate in family time. Behavior-based conflict arises when the action of one role contradicts the expectations of another. Houlfort et al. [6] emphasized that a multifaceted understanding of WFC offers depth and detail to a notion traditionally viewed as merely a work domain interfering with the family domain.

Among the three different forms of WFC, however, only time-based and strain-based WFC are the most frequently measured and investigated conflict dimension [6,16]. Therefore, in this study, we use time-based and strain-based WFC.

### 1.3. Emotional Exhaustion (EE)

Maslach [32] argued that emotional exhaustion is a component of burnout, a psychological syndrome. Cordes and Dougherty [33] defined emotional exhaustion as a loss of energy and the perception that one’s emotional resources have been depleted. Emotional exhaustion is one of the three dimensions of burnout that are regarded as the primary cause of burnout [34,35]. The first dimension of burnout is emotional exhaustion. Maslach [32] defines emotional exhaustion as a state in which workers feel unable to position or arrange themselves on a psychological level due to depleted emotional resources. The second dimension is depersonalization, in which individuals lose emotional attachment to customers. In other words, the personnel either ignore the customers or consider them commodities, resulting in apathy or emotional distance. The final dimension is reduced personal accomplishment, which occurs when individuals lose the desire for their employment, thereby diminishing their perceptions of competence and professional performance. In this study, we focused on the EE dimension of burnout because several prior studies revealed that EE is the central process of burnout [14,36,37]. Furthermore, Halbesleben [14] asserted that emotional exhaustion is the core of burnout while the other two aspects are responses to emotional exhaustion. In addition, EE has a stronger relationship with job-related outcomes than the other two components of burnout [38] and has been demonstrated to be the best indicator of burnout [39].

### 1.4. Passion for Work and WFC

According to the dualistic model of passion notion proposed by Vallerand and Houlfort [7], obsessive passion causes conflict with other activities. This is because of the rigorous commitment to the work that is now being performed. Individuals with OP experience conflicts when their job aspirations conflict with their family responsibilities. The spillover effect can also be used to understand the presence of WFC in individuals with OP. According to Grzywacz and Marks [40], the detrimental impact of one role might affect other roles in one’s life. Cropley and Purvis [41] revealed that individuals with an OP will continuously focus on the work they are performing. It can interfere with other roles in their life, including those related to family. As a result of the spillover effect between roles, Ratelle et al. [42] concluded that strain-based conflict occurs when individuals focus too much and carry the burden of the work they are performing. Thorgren et al. [17] conducted an identical study using the concept of off-task thoughts at work and on-task thoughts. The results revealed that those with an OP would devote their full attention to the task. This leads to a positive correlation between OP and WFC.

According to the conservation of resources (COR) concept, individuals with an OP believe that the resources they have, such as the time and energy they devote to a certain activity, interfere with their other life sphere responsibilities [43]. It implies that individuals with an OP have fewer resources. In addition, their passion is inseparable from their extremely rigorous devotion to a single activity. In addition, people with OP are more likely to experience both WIF and FIW due to an internalization process that is controlled in this way, whether as a result of pressure from within or from the external environment [6]. Furthermore, Houlfort et al. [6] revealed that OP was strongly associated with all types of WIF and FIW [6]. In the same study, Houlfort et al. [6] also emphasized that conflicts between family and work (FWC) might be caused by employees’ reluctance to accommodate family and partners’ requests for minor changes to their work schedules or plans. This is due to their unyielding dedication and passion for their work. Jung and Sohn [44] argued that individuals with obsessive passion experience intense internal pressures, struggle with activity engagement, and encounter conflict in other life domains. Consequently, those with obsessive passion experience greater negative affect and dissatisfaction with work/family-related activities [7]. They may have negative attitudes and emotions in other domains due to negative spillover effects. Furthermore, as obsessively passionate employees perceive growing demands and hurdles in their work and family roles, they tend to demonstrate a greater level of WIF and FIW in multiple role-related circumstances [16,44].

Brown et al. [45] revealed that individuals with HP are more flexible when investing resources. Caudroit et al. [16] conducted a study with a sample of elementary school teachers using both WIF and FIW. The findings demonstrated that HP had a negative effect on both types of WFC (WIF and FIW). Vallerand [9] claimed that the flexibility of investing in these resources makes it easier for individuals with HP to engage in multiple activities in different domains. In other words, HP is more effective in combining both family and work-life [6]. Houlfort et al. [6] revealed that HP was negatively associated with WIF and FIW. Harmonious passion creates a balance between numerous life roles and improves work–family enrichment, thereby motivating work–family enrichment [44]. Due to autonomous motivation, individuals with harmonious passion do not feel inclined to engage in their preferred activity; rather, they strongly commit themselves to this activity and maintain harmony with other life domains. Due to the fact that individuals with harmonious passion have a high level of self-determined motivation, they are likely to experience a high degree of autonomy in situations that demand them to synchronously perform various roles [46]. This may improve the ability to balance work and family responsibilities.

Thus, we hypothesized that OP positively affects both WIF and FIW, whereas HP negatively affects WIF and FIW

### 1.5. Passion for Work and EE

Passion is the tremendous propensity of an individual to engage in a certain desirable, self-defining behavior, which the individual values highly and is prepared to devote a great deal of time and energy to [7,10]. Chen et al. [10] argued that if an individual perceives that a job has meaning, loves the job, and is excited to continue engaging in the job, the job becomes a component of the individual’s identity, resulting in enthusiasm for the profession known as the passion for work. Despite the fact that passion promotes willing involvement, it may support both benefits and disadvantages based on the type of passion displayed [4]. Through this paradigm, researchers have developed a dualistic model of passion for work [4,7,20].

Few studies have analyzed the association between firefighters’ passion and EE. However, evidence supports the assumption that HP and OP typically have distinct motivating effects [8,10]. According to SDT, individuals’ reasons for performing their tasks are varied. SDT differentiates between controlled and autonomous involvement in a specific activity or group of activities. Lee and Cho [47] revealed that HP is associated with a variety of favorable workplace outcomes, whereas OP is associated with negative outcomes. HP may conform to the characterizations of autonomous motivation and has been associated with several positive job consequences, including well-being [48], as well as being negatively linked with burnout [48,49] and EE [10,19,21]. In the case of HP, as it is a resource that the individual actively controls, it is less likely to be lost and might even halt the loss of other resources [3,20]. Therefore, HP should decrease stress and, as a result, burnout in the form of exhaustion. In comparison, the concept of OP appears to match the notion of controlled motivation and is expected to arise in various undesirable effects, including work–family conflict [16] and EE [10,39]. Therefore, unlike HP, individuals cannot control their passion or effortlessly shift their attention to other tasks when it overwhelms them [20]. Due to OP’s overwhelming nature, there is a far greater feeling of resource loss than potential gain, which leads to additional stress [3,14,20]. Thus, we hypothesized that OP positively affects EE and HP negatively affects EE.

### 1.6. WFC and EE

According to Jawahar et al. [50], each individual has unique and limited resources. Individuals with highly demanding jobs will devote an excessive number of resources to their work. Individuals will feel deprived of resources that could be redirected to their family, resulting in WIF. Emotional exhaustion is one of the negative effects of work–family conflict as a source of stress [30]. The relationship between work–family conflict and emotional exhaustion is explained by the conservation of resources theory (COR). The term resources in COR theory refers to objects, personalities, situations, or energy possessed by individuals as a means of achieving certain life goals [51,52]. COR theory contains two fundamental premises. The first concerns acquiring resources, in which individuals increase their resource reserves by actively interacting with their surroundings. The second concerns resource conservation, in which individuals avoid situations threatening their resources. The COR theory emphasizes that it is more harmful to lose resources than to assist others in regaining lost resources. Therefore, employees will lose resources when they cannot fulfill their family life responsibilities effectively. The loss of these resources will diminish the sense of family well-being and deteriorate marital relationships, thereby becoming a major source of stress and contributing to emotional exhaustion.

How WIF and FIW can positively affect emotional exhaustion can be rationally explained by COR theory and the perspective of emotional resources. WFC tends to elicit negative emotional responses, which may necessitate emotional regulation to control and manage negative emotions. Zheng et al. [53] emphasize that this emotion regulation will then lead to the depletion of resources such as energy that can assist individuals in refocusing and performing effectively in their family and professional roles. However, due to the individual’s limited resources, the individual’s ability to manage emotional responses is also impaired [53,54]. Therefore, when someone has significant demands at work or home, it is highly likely that they will experience resource depletion, which is one of the characteristics of an emotionally exhausted individual.

Multiple studies in the area of occupational health indicate that both FIW and WIF have a positive effect on emotional exhaustion. According to Medina et al. [55], both types of WFC, namely, WIF and FIW, were positively correlated with burnout. According to the findings of Recuero and Segovia [56], WIF and FIW positively impact emotional exhaustion. In addition, their research distinguishes between two groups of respondents, namely, men and women, revealing that the effect of WIF on emotional exhaustion is greater for men. In contrast, the effect of FIW is greater for women. Similarly, the research of Zheng et al. [57] revealed that both FIW and WIF have a positive impact on burnout, but WIF has a greater impact than FIW. Yustina and Valerina [31] asserted that employees who believe they have lost their resources and are unable to meet family needs are emotionally exhausted.

Each individual has different and limited resources. When the demands of the family are very high, individuals will invest their resources excessively to carry out their responsibilities in the family. However, individuals will have difficulty carrying out their responsibilities at work because they no longer have the resources to invest in the work. In such conditions, if the individuals who experience this situation are unable to cope with the situation, their resources will be depleted, and they will experience emotional exhaustion [31,58]. Thus, WIF and FIW positively affect EE.

### 1.7. The Mediating Role of WFC

Our framework suggests that passion is a distal indicator of emotional exhaustion that initiates a causal association that activates psychological mediators (WFC). Then, the mediator contributes to the sense of emotional exhaustion. A prior study from Houlfort et al. [6] suggested that WFC provides a mediating mechanism in the relationship between passion for work and psychological distress. The study by Houlfort et al. [6] confirmed the result of Vallerand et al. [8]. Both studies suggested that OP and HP highlight different mechanisms by which passion can avoid or cause burnout. OP is linked to emotional exhaustion through its association with the conflict between one’s work and other life spheres’ domains; it appears that inflexible involvement with one’s work generated by OP results in conflict between one’s work and other life activities. Therefore, the individual cannot withdraw from the work responsibility due to their rigorous dedication to it. Consequently, the individual experiences emotional exhaustion. It was discovered that HP prevents burnout due to its negative link with conflict. Thus, HP results in a more flexible commitment to the work and prevents the individual from experiencing conflict.

In the previous section, we discussed how individuals with OP may experience an intense desire to engage in their favorite activities and have difficulty stopping themselves. Chummar et al. [46] emphasized that individuals with OP assign disproportionate significance to their own identity and that this can cause conflict with other elements of the individual’s identity and other life activities. Consequently, individuals with OP may expend valuable resources such as time and energy, rendering them unable to fully engage in other activities. Prior research indicates that individuals who are obsessively devoted to a particular activity will show significant levels of conflict with their work duties [7,8,46]. The more frequently an individual experiences FIW, the more depleted their resources become, resulting in increased EE.

In contrast, individuals with HP tend to flexibly accept their family roles as important, without any attached conditions [46]. HP encourages the adoption of a family role voluntarily and fosters a sense of autonomy in pursuing the activity [8]. Not only does HP play an important role in the experience of fluid and positive affect, but it also reduces the experience of negative affect following full engagement in a family role and creates extra resources for when the individual engages in another activity or domain, such as work [46]. Individuals with HP are able to thoroughly concentrate on the task at hand and perceive positive consequences both while and after engaging in family activities. As a result, the possibility of conflict between family and work may be reduced, potentially protecting such individuals from experiencing EE.

Thus, we hypothesized that WIF and FIW mediate the relationship between OP and EE and mediate the relationship between HP and EE.

## 2. Materials and Methods

### 2.1. Case Background

The fire service serves as the backdrop for this study. For several reasons, this context is appropriate for addressing the hypotheses. High-risk occupations such as firefighting place unique demands on employees that include regular exposure to traumatic events on an emotional and physical level as well as life-threatening scenarios with a high potential for impairment. Although firefighters have higher resilience factors than the general population [59], prior research has shown that the unique demands of firefighters’ jobs increase the risk for post-traumatic stress [1,59]; commonly, the strain is from burnout [1].

In Indonesia, each fire department has an office that serves as the location of its operational components. This office serves as a garage for firefighting vehicles, a storage facility for firefighting equipment, an information center, and a command center for the fire department. The Fire Department has two levels, starting with the Fire Department office, which represents the city/district level, and several supervising firefighter posts, which represent the sub-district level. Firefighters work in groups. Generally, there are three groups or platoons in each fire department. The division of the work schedule for each platoon is 24 h a day with 48 h off in a week. In other words, each platoon has approximately 72 h per week.

### 2.2. Participants and Procedure

This study was conducted by employing a survey of firefighters in Indonesia. We contacted fire departments in four different regions in Indonesia, comprising East Java, West Java, Central Java, and Jakarta, to participate in our survey. The survey was conducted by utilizing Google Forms, and we notified the participants that the gathered information they were asked to provide would be used purely for scientific work, that there were no right or wrong answers, and that their private details would not be revealed to guarantee their privacy and confidentiality. Among the 472 responses, only responses from married employees—those identifying as a husband or wife—were further interpreted. Previous studies indicated that adults living with a partner are more likely to perceive WFC than adults who are unmarried [16,60]. In addition, the number of children is considered a family demand that makes employees more susceptible to FWC [61]. Among the 472 responses, the usable data numbered 398 responses. The survey was conducted by sending the invitation through email, and all of the variables were assessed using a self-rating scheme. The respondents of this study consisted of males, 97.1%, and females, 2.9%. Based on Table 1, most respondents were 31–40 years old (51.4%), and tenure was 6–8 years (37.1%). The majority of our respondents were frontline firefighters (80%), senior officers constituted 19.3%, and company officers 5.7%. The majority of the respondents had two children (42.1%) and the oldest children were 5–8 years old (31%).

### 2.3. Measures

All measures used a response scale from 1 “strongly disagree” to 5 “strongly agree.” The measurements in this study were adopted from prior studies. Before it was sent to the respondents, we translated the instrument to Bahasa because the original version of the instrument was in English.

The passion for work was measured using the items from Vallerand and Houlfort [7], which consists of two dimensions, namely, HP and OP, wherein each dimension has six items. Work–family conflict was measured using the items from Houlfort et al. [6], which consist of WIF and FIW. Each WFC dimension has two sub-dimensions: time-based (three items) and strain-based (three items). Emotional exhaustion was measured using the Maslach Burnout Inventory (MBI) [32]. Emotional exhaustion has nine items of the MBI.

### 2.4. Data Analysis

To test our proposed hypotheses, we employed SmartPLS 3.3.6 by SmartPLS GmbH, Hamburg, Germany. PLS was used because it enables researchers to deal with complex structural models with a large number of constructs, indicators, and mediation mechanisms [62]. The procedure for conducting SmartPLS consists of two steps. First, in order to justify the data quality, the measurement model needs to be assessed for validity and reliability. The validity of each measurement is evaluated by assessing the convergent and discriminant validity. The reliability assessment evaluated the score of Cronbach’s alpha and composite reliability. Second, the structural path needs to be assessed to analyze the proposed hypotheses.

## 3. Results

### 3.1. Measurement Model

To ensure the validity and reliability of our model, convergent and discriminant validity were assessed. Convergent validity is assessed by evaluating the outer loading score for each indicator and the average variance extracted (AVE) with 0.50 as the threshold value [62]. In the first round of this assessment, some indicators had low loading scores (FIW strain-based 1 and FIW time-based 1) and needed to be eliminated. After eliminating the indicators with low loading scores, our measurement model met with the requirement of discriminant validity. Table 2 shows the loading factor scores and the AVE.

Second, the discriminant validity was evaluated using both the Fornell–larcker criterion and heterotrait–monotrait (HTMT) correlations, as per the suggestion by Henseler et al. [63]. An HTMT score greater than 0.9 implies a lack of discriminant validity [62,63]. As shown in Table 3, all the HTMT scores were below 0.90, implying that the variables in this study were discriminant. Last, the consistency of the measurements was evaluated by checking the Cronbach’s alpha and composite reliability scores. If the score exceeds 0.70, it is considered reliable [62]. Thus, we can conclude that both convergent and discriminant validity were fulfilled in this study.

### 3.2. Hypothesis Testing

#### 3.2.1. Direct Effect

After assessing the data quality, the next step is testing the proposed hypotheses. To test the proposed hypotheses, we followed a suggestion from Sulistiawan et al. [64] by conducting a bootstrap with 5000 subsamples to assess the strength of the relationship. The results of the hypothesis-testing procedure are summarized in Table 4. According to hypothesis 1, OP positively affects WIF. In Table 4, OP positively affects WIF (β = 0.582, *p* < 0.001); thus, hypothesis 1 was supported. Hypothesis 2 proposed that OP has a direct influence on FIW. Based on Table 4, OP positively affects FIW (β = 0.446, *p* < 0.001); thus, hypothesis 2 was supported. As illustrated in Table 4, HP negatively affects WIF and FIW, thus supporting hypothesis 3 and hypothesis 4 (β = −0.445, *p* < 0.001; β = −0.429, *p* < 0.001). According to hypothesis 5, there is a positive direct relationship between OP and EE, and our result does not confirm the hypothesis. Our result revealed that OP negatively affects EE. Thus, hypothesis 5 was not supported (β = −0.071, *p* < 0.05). On the other hand, hypothesis 6 was supported because HP negatively affects EE (β = −0.110, *p* < 0.01). Our results also confirm that both WIF and FIW positively affect EE, thus confirming hypothesis 7 (β = 0.485, *p* < 0.001) and hypothesis 8 (β = 0.413, *p* < 0.001)

#### 3.2.2. Indirect Effect

To analyze the indirect effect, Zhao et al.’s [65] methodology was utilized. According to Zhao et al. [65], when the direct and indirect effects are significant, the mediating effect is only partial. When the direct effect is insignificant, but the indirect effect is significant, the mediating effect is considered fully mediated. The results regarding the indirect effect are summarized in Table 5. Table 5 indicates that the indirect effect of OP on EE through WIF and FIW is significant (β = 0.242, *p* < 0.001; β = 0.215, *p* < 0.001), thus confirming hypothesis 9 and hypothesis 10. Since the direct effect of OP on EE is not significant, the mediating effect of WIF and FIW in this relationship is considered fully mediated. The indirect effect of HP on EE through WIF and FIW is also significant (β = −0.207, *p* < 0.001; β = −0.185, *p* < 0.001); thus, hypothesis 11 and hypothesis 12 were supported. FIW and WIF are considered to partially mediate the relationship between HP to EE because the direct effect of HP on EE is significant.

The predictive accuracy of the model was evaluated using the suggestion from Cohen [66], specifically with respect to the coefficients of determination (R^2^), which are 0.02 (weak), 0.13 (average), and 0.26 (strong). The model demonstrated high prediction accuracy with R^2^ scores of 0.347 for WIF, 0.215 for FIW, and 0.761 for EE. In addition, the model also indicated good predictive validity, with Q^2^ scores ranging from 0.196 (WIF) to 0.597 (EE), and all were substantially above zero [62].

## 4. Discussion

This study aims to advance our understanding of the relationship between passion for work, work–family conflict, and employees’ emotional exhaustion in several ways: first, by analyzing the relationship between passion for work and WFC, employing the direction of WFC and dimensions of the conflict; second, by assessing the mediating effect of WFC, which consists of WIF and FIW, in the relationship between passion for work and emotional exhaustion.

The results of this study revealed that both HP and OP significantly influence WIF and FIW in different ways. Our results confirm that OP facilitates the WIF and FIW. This suggests that firefighters with an OP tend to invest their time, energy, and effort in their jobs, which will impede their level of family involvement. It will increase their degree of excessive involvement in the workplace, leaving them with less energy and time to devote to their families. The results of this study corroborate the findings of previous studies indicating that people with OP tend to be unable to separate themselves from the passion and strict perseverance they generate and that workers with OP are more likely to be resource-constrained and thus more susceptible to experiencing WFC [6,9]. People with OP tend to be excessively committed and engaged in an activity. Consequently, they experience conflict in their various roles or domains in life. People with OP admit that an excessive commitment to a work or family activity will prevent them from investing time, energy, and effort in other activities. The findings of this study confirm that people with OP tend to have difficulty balancing their roles, making them susceptible to WIF and FIW

In contrast, firefighters with HP are generally able to balance work and family responsibilities, making them less vulnerable than those with OP. This result is consistent with the COR concept wherein those with HP tend to have more resources, allowing them to manage demands from multiple domains. In addition, those with HP are less likely to lose resources than those with OP. These findings validate previous research indicating that HP is characterized by flexibility and mindfulness at work, which allow individuals to engage in other activities, such as family activities [9]. In addition, this finding can be explained by the autonomous internalization in DMP. HP is the result of a continuous and discretionary internalization process for a specific activity, without external pressure to engage in the activity until it becomes an integrated part of one’s self-identity. HP can generate strong motivation to engage in voluntary activities that are enjoyed and ingrained in the individual’s character [20].

Our results also confirm that both WIF and FIW positively affect EE. This result corroborates previous studies [12,57,67]. There is a correlation between high job demand and emotional exhaustion, which may contribute to WIF among firefighters. This confirms previous research indicating that alternating day shifts are more likely to cause conflicts between work and family responsibilities [14]. The less time and energy firefighters have to meet these demands, the more emotionally exhausted they will become.

Interestingly, contrary to our hypothesis, the result of this study revealed that OP has a negative effect on EE. The negative impact of OP on EE addresses the question of whether OP is always detrimental. The influence of collectivist culture in Indonesia is one factor that helps to explain this phenomenon. Indonesia has a collectivity culture score of 86, which is extremely high compared to western countries [68]. A collectivist culture causes OP to have a negative effect on EE, as it is associated with interdependent self-construal based on the expectations of communities, receiving assistance from others to overcome problems, and family support that helps improve well-being despite encountering difficulties [5]. These findings suggest that OP can be associated with a double-edged sword because, on the one hand, OP can increase WFC while, on the other hand, OP decreases EE. As a result, a counterintuitive impact emerges wherein people experience tension as a consequence of OP. Still, the benefits, including a positive self-image that extends to positive appraisals of well-being, diminish the negative outcomes [5,69]. Firefighters may also view their line of work as difficult and gratifying, and they may view the consequences of an obsessively passionate zeal as a necessary price to pay for success in their profession.

Our result confirms that WIF and FIW mediate the relationship between passion for work and EE. It implies that the conflict between one’s work and other life activities contributes to emotional exhaustion as a consequence of the interaction between a person’s enthusiasm for their work and this individual’s passion. Therefore, a rigorous commitment to one’s task prompted by an OP hinders one from experiencing work satisfaction and creates conflict between work and family. This study confirms prior burnout studies that hypothesized that because of a person’s passion, they stay heavily engaged in their work, are still unable to disengage from it, and, under extreme conditions, will develop emotional exhaustion [8,70,71]. The findings indicated that with HP, as opposed to suffering emotional exhaustion, individuals flourish in their careers. Specifically, it was discovered that HP prevents emotional exhaustion due to its negative link with conflict.

### 4.1. Practical Implications

According to the findings of these studies, firefighters’ passion for their work can shield them against WFC and, in turn, psychological suffering. Nevertheless, not all passions are identical. Therefore, organizations should encourage the harmonious development of employees’ work-related passion. Generally, the results suggested adjustable and conscientious participation in one’s working life, allowing passionate persons to participate and engage in other activities. In addition, organizations may start providing employees with the skills necessary to deal with challenging work-related situations and distracting thinking patterns and, for instance, enhancing employees’ mindfulness abilities as a cognitive-emotional method for sharing their individual lives. The findings indicate that WIF has a stronger influence on emotional outcomes than FIW. The results have demonstrated the stressful nature of working as a firefighter, which entails long working hours, intense job stressors, and complex tasks. The high job stress firefighters face tends to affect their personal lives, depletes their emotional resources, and negatively impacts their well-being. WIF specifically enhances the level of emotional exhaustion more than FIW. Thus, the implementation of treatments or practices are suggested to minimize job stress and WIF, such as providing work and family-related support to reduce the spillover between work and family domains. Supervisors should provide such a family-supportive policy or behavior by firstly providing emotional support, such as understanding and demonstrating compassion for employees’ work–family responsibilities. Secondly, supervisors should provide instrumental assistance, such as meeting the needs of employees through everyday management activities, including a more flexible working arrangement to arrange shifts. Thirdly, supervisors should demonstrate how to integrate work and family through modeling conduct. Supervisors can be role models for their employees. A supervisor may demonstrate effective behaviors for balancing the work and nonwork roles. Lastly, innovative work–family management should be demonstrated in order to improve employees’ effectiveness outside of work. Supervisors can employ several possible approaches to determine innovative forms of work–family management, such as redistributing job duties to help division members work effectively in a team and soliciting input from staff to make it easier to balance work and family responsibilities.

### 4.2. Limitations and Future Research

Despite its significant contributions to managerial and theoretical applications, our study has several limitations. First, the present study employed a cross-sectional study; it cannot generalize the results regarding the relationship between the variables. Therefore, we suggest that longitudinal studies could be utilized for future studies. Second, this research relied exclusively on self-reported information. Future studies should try to replicate the current findings by collecting the data from the informant’s evaluation of WFC and emotional exhaustion (e.g., spouse). Third, the present study was conducted in the context of firefighters only. Thus, in order to enhance the generalizability of the present study, future research may expand the context of the present study. Fourth, potential moderating variables should be considered for future studies. Future studies may employ moderating variables (e.g., family and coworker support) that could act as a buffer in minimizing the effect of OP on work–family conflict.

## 5. Conclusions

This study aimed to investigate how passion for work affects emotional exhaustion in the context of firefighters. Specifically, we proposed the use of the dualistic model of passion and conservation of resources theory in developing our proposed hypotheses. The dualistic model of passion, which consists of HP and OP, accounts for different firefighters’ work and family lives. Additionally, we analyzed the mediating effect of work–family conflict in the relationship between passion for work and emotional exhaustion. We evaluated our proposed hypotheses by utilizing PLS-SEM and assessing the validity and reliability of each construct. Our study contributes in two ways to passion-for-work research. It contributes to the DMP by demonstrating the consequences of harmonious and obsessive passion. In addition, the present study expands our understanding of the effects of two types of passions in a non-Western country and a particular occupation—firefighters. Second, the contradictory findings of prior studies necessitate additional research to determine whether harmonious passion always results in positive outcomes and whether obsessive passion results in negative outcomes. Our findings partially validate the positive outcomes of harmonious passion and the negative outcomes of obsessive passion. Harmonious and obsessive passions decrease both the WIF and FIW directions of WFC. Additionally, harmonious passion decreases EE. Interestingly, obsessive passion also reduces EE. This result contributes to the body of knowledge on passion for work by demonstrating that obsessive passion is not always detrimental.

## Figures and Tables

**Table 1 ijerph-19-14629-t001:** Respondents Characteristics.

Demographic Characteristics	Classifications	Frequency	%
Gender	Male	386	97.10%
	Female	12	2.90%
Age	21–30	119	30%
	31–40	205	51.40%
	above 40	74	18.60%
Job Position	Senior officers	77	19.30%
	Company officers	23	5.70%
	Frontline firefighters	299	75.00%
Tenure	Less than three years	45	11.40%
	3–6 years	123	30.90%
	6–8 years	178	44.72%
Number of Children	0	40	10%
	1	134	33.6%
	2	167	42.1%
	More than 2	57	14.3%
Age of the Oldest Children	Less than five years	111	28%
	5–8 years	125	31%
	8–11 years	94	24%
	Above 11 years	68	17%

**Table 2 ijerph-19-14629-t002:** The validity and reliability results.

Variables’	Indicators	Mean	Standard Deviation	Loading Factor	Cronbach’s Alpha	CR	AVE
Obsessive Passion	OP1	3.410	0.804	0.823	0.864	0.892	0.581
	OP2			0.731			
	OP3			0.763			
	OP4			0.631			
	OP5			0.834			
	OP6			0.775			
Harmonious Passion	HP1	4.051	0.753	0.907	0.905	0.924	0.673
	HP2			0.937			
	HP3			0.792			
	HP4			0.704			
	HP5			0.686			
	HP6			0.863			
Work interferes with family	WIF_S1	2.350	0.846	0.884	0.932	0.947	0.748
	WIF_S2			0.881			
	WIF_S3			0.869			
	WIF_T1			0.780			
	WIF_T2			0.877			
	WIF_T3			0.895			
Family interferes with work	FIW_S1	2.290	0.776	0.908	0.922	0.945	0.811
	FIW_S2			0.937			
	FIW_T1			0.863			
	FIW_T2			0.892			
Emotional Exhaustion	EE1	2.137	0.831	0.893	0.967	0.972	0.793
	EE2			0.865			
	EE3			0.928			
	EE4			0.899			
	EE5			0.913			
	EE6			0.891			
	EE7			0.833			
	EE8			0.884			
	EE9			0.904			

Notes: CR: Composite Reliability; AVE: Average Variance Extracted.

**Table 3 ijerph-19-14629-t003:** Fornell–larcker and HTMT results.

	1	2	3	4	5
OP	**0.762**	*0.502*	*0.429*	*0.281*	*0.204*
HP	0.356	**0.821**	*0.222*	*0.260*	*0.352*
WIF	0.423	−0.239	**0.865**	*0.889*	*0.850*
FIW	0.294	−0.271	0.826	**0.900**	*0.882*
EE	0.207	−0.365	0.810	0.835	**0.890**

Notes: OP—Obsessive passion; HP—Harmonious Passion; WIF—Work interferes with family; FIW—Family interferes with work; EE—Emotional Exhaustion. The diagonal values in **bold** are the square root of AVE. The values in *italic* above the diagonal are the HTMT score. The values on the lower diagonal are latent variable correlations.

**Table 4 ijerph-19-14629-t004:** Results of direct effect.

	β	T Statistics	*p* Values	5.00%	95.00%	Result
H1: OP → WIF	0.582	12.758	0.000	0.490	0.669	Supported
H2: OP → FIW	0.446	9.453	0.000	0.349	0.528	Supported
H3: HP → WIF	−0.445	10.065	0.000	−0.527	−0.356	Supported
H4: HP → FIW	−0.429	9.637	0.000	−0.517	−0.346	Supported
H5: OP → EE	−0.071	1.831	0.043	−0.138	−0.009	Not supported
H6: HP→EE	−0.110	3.172	0.001	−0.156	−0.067	Supported
H7: WIF → EE	0.485	5.183	0.000	0.256	0.565	Supported
H8: FIW → EE	0.413	5.782	0.000	0.322	0.645	Supported

Notes: OP—Obsessive passion; HP—Harmonious Passion; WIF—Work interferes with family; FIW—Family interferes with work; EE—Emotional Exhaustion.

**Table 5 ijerph-19-14629-t005:** Results of the indirect effect.

	β	T Statistics	*p* Values	5.00%	95.00%	Result
H9: OP -> WIF -> EE	0.242	4.621	0.000	0.143	0.348	Supported
H10: OP -> FIW -> EE	0.215	4.698	0.000	0.129	0.307	Supported
H11:HP -> WIF -> EE	−0.185	4.460	0.000	−0.269	−0.11	Supported
H12: HP -> FIW -> EE	−0.207	4.828	0.000	−0.298	−0.133	Supported

Notes: OP—Obsessive passion; HP—Harmonious Passion; WIF—Work interferes with family; FIW—Family interferes with work; EE—Emotional Exhaustion.

## Data Availability

Not applicable.

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
