# Peer review of "Linking Passion for Work and Emotional Exhaustion in Indonesian Firefighters: The Role of Work–Family Conflict"

_ijerph, 2022, doi:10.3390/ijerph192214629_

Round 1

Reviewer 1 Report (Previous Reviewer 1)

Wow, I really believe this paper is better with your revisions.  It is much easier to understand and how you arrive at your conclusions is very clear. This is a good study, I still believe the generalizability is a bit difficult, but the study is solid and I enjoyed reading it and believe it adds to the scientific literature.

This manuscript is a resubmission of an earlier submission. The following is a list of the peer review reports and author responses from that submission.

Round 1

Reviewer 1 Report

I believe your research is important and work-stress is definitely an issue in many cultures.  The design of the study was detailed and thought out, but found the result in a collectivist culture of  the negative correlation between the OP and EE is the most important result.  It makes the generalizability of the study poor.  Choosing firefighters where they spend nights away from home every week also decreases the generalizability of the study for most families.  In a majority of countries and cultures, parents/spouses work their shift each day and come home. The disruption of time at home may be felt more intensely in these cases.  I thought your study was written well, it was a bit confusing because so many acronyms were used, but that is easily remedied.  I believe that you were very upfront regarding the challenges and limitations you faced, and I will recommend it be published.  I just do not think anything very novel or different was discovered with this research.  

Author Response

Thank you for your comments, please see the attachment.

Reviewer 2 Report

General comments

The aim of this manuscript is to examine the relationship between passion for work and emo-tional exhaustion with work-family conflict as mediator. This is a cross-sectional study in-tended to support the dualistic model of passion and to replicate previous findings on these relationships with firefighters. The purpose of the study is interesting because it helps understanding how passion for work can lead to emotional exhaustion. Nevertheless, some questions remain.

1) The authors choose emotional exhaustion rather than burnout. Why? Is there a rationale for that? Moreover, causes of emotional exhaustion should be more detailed by using the COR model in that sense. Why emotional exhaustion is a problem in firefighters? This must be described a bit more. As the authors said, emotional exhaustion in firefighters could be associated with other causes such as physically demanding work, time constraints, and emotionally taxing events. In addition, what mean “losing resources” for the authors?

2) In the introduction section, the section on “Passion and WFC” is not clear to me and the authors talk about FWC also. Therefore, I recommend rewriting the introduction section by shortened it and focusing on the most important information. Emotional exhaustion is the main purpose of the study and should be discussed earlier in the introduction section. Overall, the information is scattered in the introduction, and it becomes difficult to get a precise idea of the relationships between all the variables.

3) The authors should be less ambitious concerning the purpose of their study. A cross-sectional study cannot claim to reveal effects. Therefore, the objective should be rewritten by considering the methodology of the study. The authors don’t investigate the causal effect of passion for work on emotional exhaustion. Moreover, they specify it within the limits of their study with good reason. In addition, they propose 12 hypotheses which is too much for a correlational study. The paragraph explaining the negative relationship between OP and EE needs further explanation. Overall, the authors should explain more clearly the purpose of their study and why they choose a cross-sectional study for that.

4) The authors should distinguish obsessive and harmonious passion more in depth. Moreover, the authors talk about the Houlfort study by revealing a positive relationship between harmonious passion and time-based family-work conflict. What do they do to this information?

5) The results section should be written differently as there is a lot of redundancy. 

6) In the conclusion section, the authors wrote that the study gives a more complete picture of the relationship between passion and work-family conflict (WFC). What exactly does this study add to what already exists? In addition, in the discussion section, the authors should discuss the vicious circle of emotional exhaustion based on their findings.

Author Response

(The authors gave the same response as above.)
